# The association of bearing surface materials with the risk of revision following primary total hip replacement: A cohort analysis of 1,026,481 hip replacements from the National Joint Registry

**Michael R. Whitehouse**[1,2☺]**, Rita Patel**[1☺]**, Jonathan M. R. French**[1]**, Andrew D. Beswick**[1,2]**, Patricia Navvuga**[1]**, Elsa M. R. Marques**[1]**, Ashley W. Blom**[1,2,3‡]**, Erik Lenguerrand**[1,2¤‡*]

1 University of Bristol Medical School, Translational Health Sciences, Musculoskeletal Research Unit, Bristol, United Kingdom, 2 National Institute for Health Research Bristol Biomedical Research Centre, University Hospitals Bristol and Weston NHS Foundation Trust and University of Bristol, Bristol, United Kingdom, 3 Faculty of Health, University of Sheffield, Sheffield, United Kingdom

☺ These authors contributed equally to this work.
¤ Current address: Translational Health Sciences, Bristol Medical School, University of Bristol, Bristol, United Kingdom
‡ AWB and EL also contributed equally to this work.
* erik.lenguerrand@bristol.ac.uk

**Data Availability Statement:** The research project was reviewed and approved by the HQIP-National

## Abstract

### Background

The risk of re-operation, otherwise known as revision, following primary hip replacement depends in part on the prosthesis implant materials used. Current performance evidences are based on a broad categorisation grouping together different materials with potentially varying revision risks.

We investigated the revision rate of primary total hip replacement (THR) reported in the National Joint Registry by specific types of bearing surfaces used.

### Methods and findings

We analysed THR procedures across all orthopaedic units in England and Wales. All patients who received a primary THR between 2003 and 2019 in the public and private sectors were included. We investigated the all-cause and indication-specific risks of revision using flexible parametric survival analyses to estimate adjusted hazard ratios (HRs). We identified primary THRs with heads and monobloc cups or modular acetabular component THRs with head and shell/liner combinations.

A total of 1,026,481 primary THRs were analysed (Monobloc: $n = 378,979$ and Modular: $n = 647,502$) with 20,869 (2%) of these primary THRs subsequently undergoing a revision episode (Monobloc: $n = 7,381$ and Modular: $n = 13,488$).

Joint Registry board and the data were analysed in a data safe haven approved by this data provider. The studied data and related statistical codes are archived on the University of Bristol network. The related files remains accessible to all those with due HQIP-National Joint Registry permissions that are beyond our control. Further information provided by the data controller can be accessed here:https://www.njrcentre.org.uk/research/research-requests/ contains information on research data access request to the National Joint Registry. The authors will share all codes and data with those who hold permission from the National Joint Registry, as per this data controller requirements.

**Funding:** This article presents independent research funded by CeramTec GmbH, Plochingen (https://www.ceramtec-group.com/en/) (MRW, RP, JF, PN, EM, AWB and EL), the NIHR Bristol Biomedical Research Centre (https://www.bristolbrc.nihr.ac.uk/) (MRW, AB, AWB and EL) and the NIHR Comprehensive Clinical Research Network (https://www.nihr.ac.uk/explore-nihr/support/clinical-research-network.htm) (MRW). The funders had no role in study design, data collection and analysis, decision to publish, or preparation of the manuscript.

**Competing interests:** MRW, AB, EM, AWB and EL are applicants on research grants funded by the NIHR to the University of Bristol. MRW, EM, AWB and EL hold a contract with the National Joint Registry (FTS 010307-2022: Statistical Analysis, Support and Associated Services). MRW conducts teaching on basic sciences for FRCS candidates, his employer receives market rate payments for his time for this teaching from Heraeus GMbH. The other authors have declared that no competing interests exist.

**Abbreviations:** AIC, Akaike information criterion; ASA, American Society of Anesthesiologists; BIC, Bayesian information criterion; BMI, body mass index; CI, confidence interval; ECC, Ethics and Confidentiality Committee; HCLPE, highly crosslinked polyethylene; HR, hazard ratio; RCT, randomised controlled trial; THR, total hip replacement.

For monobloc implants, compared to implants with a cobalt chrome head and highly crosslinked polyethylene (HCLPE) cup, the all-cause risk of revision for monobloc acetabular implant was higher for patients with cobalt chrome (hazard rate at 10 years after surgery: 1.28 95% confidence intervals [1.10, 1.48]) or stainless steel head (1.18 [1.02, 1.36]) and non-HCLPE cup. The risk of revision was lower for patients with a delta ceramic head and HCLPE cup implant, at any postoperative period (1.18 [1.02, 1.36]).

For modular implants, compared to patients with a cobalt chrome head and HCLPE liner primary THR, the all-cause risk of revision for modular acetabular implant varied non-constantly. THRs with a delta ceramic (0.79 [0.73, 0.85]) or oxidised zirconium (0.65 [0.55, 0.77]) head and HCLPE liner had a lower risk of revision throughout the entire postoperative period.

Similar results were found when investigating the indication-specific risks of revision for both the monobloc and modular acetabular implants.

While this large, nonselective analysis is the first to adjust for numerous characteristics collected in the registry, residual confounding cannot be rule out.

## Conclusions

Prosthesis revision is influenced by the prosthesis materials used in the primary procedure with the lowest risk for implants with delta ceramic or oxidised zirconium head and an HCLPE liner/cup. Further work is required to determine the association of implant bearing materials with the risk of rehospitalisation, re-operation other than revision, mortality, and the cost-effectiveness of these materials.

## Author summary

### Why was this study done?

- The classifications used to categorise hip implants in national registries are typically broad and may not allow interested parties to fully understand the risk of postoperative revision (i.e., need for further surgery) associated with different types of implant materials.

- This research aimed to report the risk of revision by the detailed implant materials used to help the surgical community, and therefore patients, identify those with the lowest risk of further surgery or revision which will improve shared decision-making prior to surgery.

### What did the researchers do and find?

- This research analysed 1,026,481 primary total hip replacements (THRs) performed in England and Wales with information up to 15 years after these initial hip replacement operations.

- Hip prostheses with a delta ceramic or oxidised zirconium head and highly crosslinked polyethylene liner or cup had the lowest risk of revision throughout the 15 years following surgery.

- These findings were also found when investigating the specific reasons for revision hip replacements being performed.

### What do these findings mean?

- These results, from one of the largest registries in the world covering all public and private health care structures in England and Wales will help hospitals, surgeons and therefore patients to choose hip implants and combinations of them that can be used with the lowest risk of revision.

- These results are not from a randomised controlled trial and therefore it is impossible to control for all factors that can influence the risk of revision.

## Introduction

Total hip replacement (THR) is widely used to treat diseased and damaged joints with over 100,000 performed annually in the United Kingdom [1,2]. Although 58% to 78% of THRs last more than 25 years [3], many still fail resulting in 5,073 revisions annually in the UK in 2021 [1,4]. People experience worse pain and function after revision compared with primary THRs and often require further revision [5,6]. Each revised THR lasts about half as long as its predecessor [7]. The most common reasons for revision are aseptic loosening, dislocation, periprosthetic fracture, infection, adverse soft tissue reaction to wear debris and pain [1]. These causes are not mutually exclusive and in many cases are inextricably linked. For example: wear particles activate macrophages that have been implicated in initiating loosening; wear of the prostheses can lead to instability; and particulate debris from wear damages tissues and results in an environment prone to infection [8,9].

The role of fixation [10], instability [11], and infection [12], in implant longevity have been extensively studied from an epidemiological view point. Even though the tribological mechanisms of wear are well understood [13], large scale, representative epidemiological studies conducted across multiple settings, or whole health care system assessing the association between different bearing surfaces and failure remain sparse [14].

The materials used in the bearing surfaces are typically described by the material that makes up the articulating surface of the femoral head followed by the material that makes up the articulating surface of the cup or liner of the acetabular shell.

Studies usually use broad grouping (metal-on-polyethylene, ceramic-on-polyethylene, ceramic-on-ceramic, metal-on-metal, ceramic-on-metal) when comparing bearing materials despite these groupings consisting of aggregations of different materials. For example, metal heads are commonly either stainless steel or cobalt chrome, polyethylene has evolved over time with highly crosslinked polyethylene (HCLPE) now in widespread use and there are different types of ceramic such as alumina and delta ceramics. Some national joint replacement registries report good performance for hip implants with ceramic acetabular component [15–

19] but these results are unadjusted and not underpinned by formal statistical tests. Reporting from one registry, adjusting for year of primary surgery, patient age, sex, and surgical year, have not shown statistical different results for hip implants with ceramic cup [20].

No study, using nationally representative data, has yet provided in depth evidence on the performance of each specific hip bearing surface materials, underpinned by adjusted modelling and throughout the whole post-primary operation period. It is therefore still unclear which of the materials implanted in routine orthopaedic care are the most effective options; hence, the wide variety of practice and changing patterns of practice observed.

To obtain representative evidence on the risk of revision associated with the specific type of bearing surfaces used for primary THR, we investigated the revision rates as reported in the National Joint Registry, across all orthopaedic units in England and Wales between 2003 and 2019.

## Methods

### Ethics statement

With support under Section 251 of the National Health Service (NHS) Act 2006, the Ethics and Confidentiality Committee (ECC) (now the Health Research Authority Confidentiality Advisory Group) allows the NJR to collect patient data where consent is indicated as "Not Recorded."

Before Personal Data and Sensitive Personal Data are recorded, express written patient consent is provided. The NJR records patient consent as either "Yes," "No," or "Not Recorded."

### Data source

We assessed data from the NJR—established in 2003. It currently records all primary and revision hip replacements done in hospitals in England, Wales, Northern Ireland, the Isle of Man and the States of Guernsey. A total of 1,204,423 primary THR procedures had been recorded in England and Wales until 31 December 2019. Our analysis is based on 1,027,098 (85.3%) primary procedures, recorded in the NJR, which include patient consent and identifiers that allow revisions to be linked to primary operations with an identifiable head-cup or head-liner combination. Resurfacing procedures, stemmed MoM THR procedures, procedures with bearing implant materials that could not be resolved, with a dual-mobility bearing, implants with a monobloc acetabular component with a cup made of a single material other than HCLPE or non-HCLPE, a modular acetabular component with cobalt chrome liner or a rare combination (i.e., alumina head or liner with delta ceramic liner or head, or oxidised zirconium head and a non-HCLPE liner) were excluded (Fig A in S1 Text). Patients had given their consent for this study as part of their consent for data linkage in the NJR.

### Outcomes

Our analyses estimated all-cause and cause-specific revision rates. We estimated revision rates for different head-cup or head-liner combinations. Our unit of analysis for the consideration of revision outcomes is the implant (rather than patient) so we included 260,830 primary procedures performed on contralateral sides of the same patient but on different dates.

The specific reasons for revision considered were those included as indications for revision listed on the NJR Minimum Data Set forms and categorised here as aseptic loosening, peri-prosthetic fracture, implant wear, malalignment, dislocation or subluxation, pain, infection, and any other reasons (lysis, implant fracture, head-socket mismatch, adverse soft tissue reaction(s), other indications).

### Exposure and adjustment factors

Component data was ascertained from the implants used at the time of primary surgery and uploaded to the NJR. Materials were defined by catalogue numbers and review of implant data publicly available from manufacturers. We first identified the primary THRs with heads and monobloc cups. We modelled the modular acetabular component THRs with a head and shell/liner combination separately. Preassembled acetabular implants (e.g., a metal shell with a ceramic liner that is provided to the surgeon as a preassembled implant identified by a single catalogue number) are modelled with the modular acetabular group. A full description of the different types of head, cup, shell, and liner materials is provided in Table 1. THRs with an unclear implant construct were excluded from this report. Implants with a cobalt chrome head and an HCPLE liner or cup were used as the reference group as this was the most commonly used implant bearing surface combination over the studied period. The combination remains among the most popular across numerous national health organisations including the NHS and the care organisations in the Nordic countries [15,20].

We also accounted for the year of the primary surgery completion, the materials used in the stem, shell, head size, implant component fixation, patient gender, age at the primary procedure, body mass index (BMI), and American Society of Anesthesiologists (ASA) grade. These variables are recorded by the surgeon or their delegate on the NJR Minimum Data Set forms [21].

### Statistical analysis

We used flexible parametric survival models that estimate hazard ratios (HRs) by bearing materials assuming that they were likely to vary over time [22]. Restricted cubic splines were used to model the baseline hazard function and the time-dependent effects associated with the bearing materials combination. For each model, the best fitting model with the most parsimonious number of knots was determined using the smallest Akaike information criterion (AIC) and Bayesian information criterion (BIC) [23]. This research focussed specifically on implant-survivorship, i.e., net implant failure; therefore, non-competing rather than competing-risk modelling was used [24]. The analyses were adjusted for year of primary surgery, patient age, gender, BMI, ASA grade, implant fixation, head component size, and stem materials. We also assessed HRs for each specific reason for revision described above.

All analyses were designed prior to analysis. They were amended a posteriori to further adjust for year of surgery, to identify whether how potential changes in clinical practices over time influenced results. The adjusted results, with or without this adjustment were similar. An additional a posteriori analysis was conducted to compare modular implants with the lowest risks of revision. The same modelling strategy was used.

This study is reported as per the "Strengthening the Reporting of Observational Studies in Epidemiology (STROBE): guidelines for reporting observational studies" statement (see S1 Strobe Checklist).

## Results

A total of 1,026,481 out of 1,204,423 primary total hip replacements are included in the analysis (Monobloc: $n$ = 378,979 and Modular: $n$ = 647,502) (Fig A in S1 Text) with 20,869 (2%) procedures subsequently undergoing a linked first revision episode (Monobloc: $n$ = 7,381 and Modular: $n$ = 13,488) (Tables 1 and A–H in S1 Text).

**Table 1. Description of the studied sample and all-cause revision rate.**

| | | Monobloc | | | Modular | | |
|---|---|---|---|---|---|---|---|
| | | No. | Revision(n) | Rate (per 10,000) | No. | Revision(n) | Revision (per 10,000) |
| **Year of primary surgery** | **<2010** | 111,781 | 2,481 | 222.0 | 111,028 | 4,539 | 408.8 |
| | **2010–2014** | 127,413 | 2,939 | 230.7 | 228,155 | 5,378 | 235.7 |
| | **2015–2019** | 139,703 | 1,958 | 140.2 | 309,018 | 3,583 | 115.9 |
| **Gender** | **Female** | 251,218 | 4,410 | 175.5 | 377,914 | 7,375 | 195.1 |
| | **Male** | 127,679 | 2,968 | 232.5 | 270,287 | 6,125 | 226.6 |
| **Age at primary (years)** | **<55** | 14,368 | 446 | 310.4 | 93,190 | 2,452 | 263.1 |
| | **55 to 64** | 48,592 | 1,316 | 270.8 | 165,336 | 3,762 | 227.5 |
| | **65 to 74** | 141,877 | 3,086 | 217.5 | 231,077 | 4,420 | 191.3 |
| | **> = 75** | 174,060 | 2,530 | 145.4 | 158,598 | 2,866 | 180.7 |
| **Body mass index** | **<18.5** | 2,658 | 36 | 135.4 | 3,682 | 66 | 179.3 |
| | **[18.5, 24.9]** | 51,097 | 734 | 143.6 | 87,542 | 1,393 | 159.1 |
| | **[25, 29.9]** | 92,193 | 1,352 | 146.6 | 172,498 | 2,928 | 169.7 |
| | **>29.9** | 84,542 | 1,511 | 178.7 | 177,560 | 3,601 | 202.8 |
| | **Unknown BMI** | 148,407 | 3,745 | 252.4 | 206,919 | 5,512 | 266.4 |
| **ASA grade** | **P1—Fit and healthy** | 40,275 | 952 | 236.4 | 104,981 | 2,322 | 221.2 |
| | **P2—Mild disease not incapacitating** | 259,297 | 5,007 | 193.1 | 446,785 | 9,018 | 201.8 |
| | **P3—Incapacitating systemic disease** | 76,495 | 1,373 | 179.5 | 93,486 | 2,098 | 224.4 |
| | **P4—Life threatening disease** | 2,783 | 46 | 165.3 | 2,901 | 61 | 210.3 |
| | **P5—Expected to die within 24 h with or without an operation** | 47 | 0 | 0.0 | 48 | 1 | 208.3 |
| **Fixation type** | **Cemented** | 345,036 | 6,627 | 192.1 | 674 | 21 | 311.6 |
| | **Hybrid** | 9 | 0 | 0.0 | 242,042 | 4,098 | 169.3 |
| | **Reverse hybrid** | 30,394 | 638 | 209.9 | 89 | 8 | 898.9 |
| | **Uncemented** | 35 | 3 | 857.1 | 395,493 | 9,052 | 228.9 |
| | **Unclassified** | 3,423 | 110 | 321.4 | 9,903 | 321 | 324.1 |
| **Stem composition** | **Stainless steel** | 289,364 | 5,219 | 180.4 | 178,564 | 2,785 | 156.0 |
| | **Cobalt chrome** | 54,909 | 1,371 | 249.7 | 64,626 | 1,384 | 214.2 |
| | **Titanium** | 30,680 | 656 | 213.8 | 391,672 | 8,865 | 226.3 |
| | **Other or unknown** | 3,944 | 132 | 334.7 | 13,339 | 466 | 349.4 |
| **Head size (mm)** | **22.25** | 12,893 | 376 | 291.6 | 1,021 | 38 | 372.2 |
| | **26** | 1,8694 | 506 | 270.7 | 775 | 27 | 348.4 |
| | **28** | 241,452 | 4,956 | 205.3 | 144,874 | 4,296 | 296.5 |
| | **30–32** | 93,203 | 1,267 | 135.9 | 268,385 | 4,316 | 160.8 |
| | **> = 36** | 9,232 | 163 | 176.6 | 223,243 | 4,502 | 201.7 |
| | **Unknown** | 3,423 | 110 | 321.4 | 9,903 | 321 | 324.1 |
| **Head cup** | **Head: Alumina, Cup: HCLPE** | 6,621 | 82 | 123.8 | | | |
| | **Head: Alumina, Cup: non-HCLPE** | 12,255 | 273 | 222.8 | | | |
| | **Head: Cobalt Chrome, Cup: HCLPE** | 35,069 | 452 | 128.9 | | | |
| | **Head: Cobalt Chrome, Cup: non-HCLPE** | 69,075 | 1,749 | 253.2 | | | |
| | **Head: Delta Ceramic, Cup: HCLPE** | 23,279 | 261 | 112.1 | | | |
| | **Head: Delta Ceramic, Cup: non-HCLPE** | 14,064 | 285 | 202.6 | | | |
| | **Head: Stainless Steel, Cup: HCLPE** | 35,825 | 352 | 98.3 | | | |
| | **Head: Stainless Steel, Cup: non-HCLPE** | 182,709 | 3,924 | 214.8 | | | |

*(Continued)*

**Table 1.** (Continued)

| | | Monobloc | | | Modular | | |
|---|---|---|---|---|---|---|---|
| | | No. | Revision(n) | Rate (per 10,000) | No. | Revision(n) | Revision (per 10,000) |
| Shell composition | Cobalt Chrome | | | | 720 | 25 | 347.2 |
| | Stainless Steel | | | | 5,273 | 134 | 254.1 |
| | Tantalum | | | | 5,741 | 187 | 325.7 |
| | Titanium | | | | 629,205 | 13,011 | 206.8 |
| | Unknown | | | | 7,262 | 143 | 196.9 |
| Head liner | Head: Alumina, Liner: Alumina | | | | 27,602 | 907 | 328.6 |
| | Head: Alumina, Liner: HCLPE | | | | 25,191 | 405 | 160.8 |
| | Head: Alumina, Liner: non-HCLPE | | | | 3,281 | 157 | 478.5 |
| | Head: Cobalt Chrome, Liner: HCLPE | | | | 183,106 | 3,485 | 190.3 |
| | Head: Cobalt Chrome, Liner: non-HCLPE | | | | 42,389 | 1,791 | 422.5 |
| | Head: Delta Ceramic, Liner: Delta Ceramic | | | | 119,558 | 2,943 | 246.2 |
| | Head: Delta Ceramic, Liner: HCLPE | | | | 140,539 | 1,757 | 125.0 |
| | Head: Delta Ceramic, Liner: non-HCLPE | | | | 13,448 | 381 | 283.3 |
| | Head: Delta Ceramic, Pre-assembled implant | | | | 4,614 | 74 | 160.4 |
| | Head: Cobalt Chrome or Stainless Steel, Pre-assembled implant | | | | 2,648 | 69 | 260.6 |
| | Head: Oxidised, Liner: HCLPE | | | | 21,263 | 269 | 126.5 |
| | Head: Stainless Steel, Liner: HCLPE | | | | 45,672 | 574 | 125.7 |
| | Head: Stainless Steel, Liner: non-HCLPE | | | | 18,890 | 688 | 364.2 |

ASA, American Society Anaesthesiologists Physical Status Classification; HCLPE, highly crossLinked polyethylene.

## Monobloc acetabular implants (*n* = 378,979)

The risks of all-cause revision by bearing surface materials used in the implant head and monobloc cup, plotted by time elapsed since the primary procedure, are reported in Fig 1 and Table 2 (with further details in Table I in S1 Text).

Compared to implants with a cobalt chrome head and HCLPE cup (reference group), the risk of revision was lower for patients with a delta ceramic head and HCLPE cup implant, at any postoperative period. Implants with a stainless steel head and HCLPE cup had also a lower risk of revision. Implants with an alumina head and HCLPE cup had a lower risk of revision for the first 5 postoperative years. The HRs adjusted for year of primary surgery, patient gender, age, BMI, ASA physical status grade, implant fixation, shell component materials, stem component materials, and head size at 10 years compared to the reference group were 0.61 (95% CI 0.50, 0.75) for delta ceramic head and HCLPE cup; 0.75 (95% CI 0.55, 1.03) for alumina ceramic head and HCLPE cup; 0.80 (95% CI 0.66, 0.97) for stainless steel head and HCLPE cup.

In contrast, the risk of revision was higher at any postoperative time for patients with cobalt chrome heads and a non-HCLPE cup and for the first 10 post-operation years for implants with stainless steel head and non-HCLPE cup.

Similar results were found when investigating indication-specific revision (Tables J–Q and Figs B, D, F, H, J, L, N, and P in S1 Text) with higher risk of revision for implant with a non-HCLPE cup. Differences were mostly observed in the first years following the primary procedure.

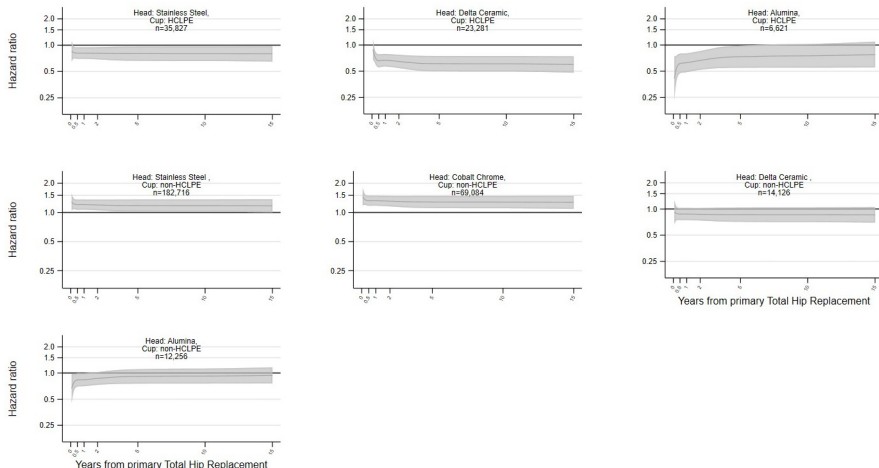

**Fig 1. Risk of revision by head and cup materials with monobloc cups (Reference: Cobalt chrome head with highly crosslinked polyethylene cup).** Flexible parametric survival model adjusted for year of primary surgery, patient gender, age, BMI, ASA grade, implant fixation, stem composition and head size. ASA, American Society of Anesthesiologists; BMI, body mass index; HCLPE, highly crosslinked polyethylene.

## Modular acetabular implants (*n* = 647,502)

Compared to patients with a cobalt chrome head and HCLPE liner primary THR (reference group), the all-cause risk of revision varied non-constantly over time (Fig 2 and Table 3 with further details in Table R in S1 Text).

THRs with a delta ceramic head and HCLPE liner had a lower risk of revision throughout the entire postoperative period. The risk of revision was also constantly lower for implants with an oxidised zirconium head and HCLPE liner and THRs with a delta ceramic head and preassembled acetabular implant. This lower revision rate was limited to the first 2 years when a delta ceramic head was paired with a delta ceramic or non-HCLPE liner. The HRs adjusted for year of primary surgery, patient gender, age, BMI, ASA physical status grade, implant fixation, shell component materials, stem component materials, and head size at 10 years compared to the reference group were 0.54 (95% CI 0.39, 0.73) for delta ceramic head pre-

**Table 2. Monobloc acetabular component-all-cause revision HR and 95% CI by time point from primary procedure-Reference: Implant with cobalt chrome head and highly crosslinked polyethylene cup.**

|  | 1 year | | | 2 years | | | 10 years | | |
|---|---|---|---|---|---|---|---|---|---|
|  | HR | 95% CI | *P*-value* | HR | 95% CI | *P*-value* | HR | 95% CI | *P*-value* |
| **Head: Stainless Steel, Cup: HCLPE** | 0.81 | [0.70, 0.95] | 0.010 | 0.81 | [0.69, 0.95] | 0.014 | 0.8 | [0.66, 0.97] | 0.030 |
| **Head: Delta Ceramic, Cup: HCLPE** | 0.67 | [0.57, 0.79] | <0.001 | 0.65 | [0.54, 0.77] | <0.001 | 0.61 | [0.50, 0.75] | <0.001 |
| **Head: Alumina, Cup: HCLPE** | 0.63 | [0.49, 0.80] | <0.001 | 0.67 | [0.52, 0.85] | 0.002 | 0.75 | [0.55, 1.03] | 0.079 |
| **Head: Stainless Steel, Cup: non-HCLPE** | 1.2 | [1.07, 1.36] | 0.005 | 1.19 | [1.05, 1.36] | 0.012 | 1.18 | [1.02, 1.36] | 0.031 |
| **Head: Cobalt Chrome, Cup: non-HCLPE** | 1.32 | [1.17, 1.49] | <0.001 | 1.31 | [1.15, 1.48] | <0.001 | 1.28 | [1.10, 1.48] | 0.002 |
| **Head: Delta Ceramic, Cup: non-HCLPE** | 0.88 | [0.75, 1.03] | 0.115 | 0.87 | [0.74, 1.03] | 0.102 | 0.86 | [0.71, 1.05] | 0.127 |
| **Head: Alumina, Cup: non-HCLPE** | 0.84 | [0.70, 0.99] | 0.05 | 0.87 | [0.73, 1.03] | 0.113 | 0.92 | [0.76, 1.13] | 0.284 |

Flexible parametric survival model adjusted for year of primary surgery, patient gender, age, BMI, American Society of Anesthesiologists grade, implant fixation, stem composition and head size. HCLPE, highly crosslinked polyethylene.

* Wald test.

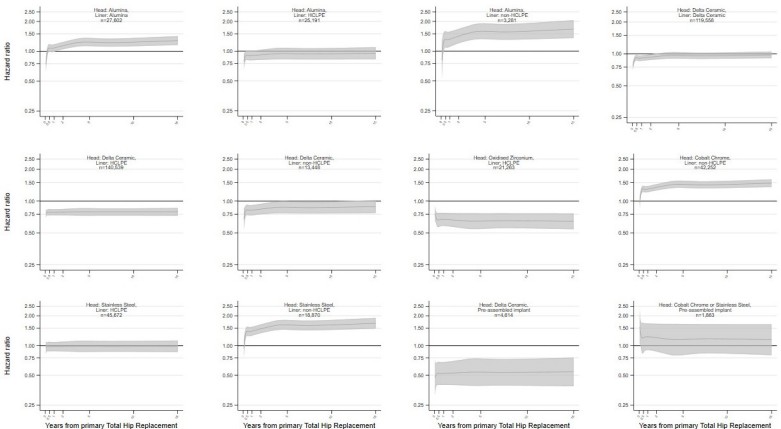

**Fig 2. Risk of revision by head and liner materials with modular cups (Reference: Cobalt chrome head with highly crosslinked polyethylene liner).** Flexible parametric survival model adjusted for year of primary surgery, patient gender, age, BMI, ASA grade, implant fixation, shell composition, stem composition, and head size. ASA, American Society of Anesthesiologists; BMI, body mass index; HCLPE, highly crosslinked polyethylene.

assembled implants; 0.65 (95% CI 0.55, 0.77) for oxidised zirconia ceramic head and HCLPE cup; 0.79 (95% CI 0.73, 0.85) for delta ceramic head and HCLPE cup.

In contrast, implants with a cobalt chrome or stainless steel head and a non-HCLPE liner had a higher risk of revision from the first months post-operation onwards. From 6 months post-operation onwards, the risk of revision was higher for implants with an alumina head and alumina, or non-HCLPE liner.

Similar results were found when investigating indication-specific revisions (Tables S–Z and Figs C, E, G, I, K, M, O, and Q in S1 Text). Compared to THRs with cobalt chrome head and

**Table 3. Modular acetabular component-All-cause revision HR and 95% CI by time point from primary procedure-Reference: Implant with cobalt chrome head and highly crosslinked polyethylene liner.**

|  | 1 year | | | 2 years | | | 10 years | | |
|---|---|---|---|---|---|---|---|---|---|
|  | HR | 95% CI | *P*-value* | HR | 95% CI | *P*-value* | HR | 95% CI | *P*-value* |
| **Head: Alumina, Liner: Alumina** | 1.09 | [1.00, 1.18] | 0.050 | 1.14 | [1.05, 1.24] | 0.003 | 1.24 | [1.13, 1.36] | <0.001 |
| **Head: Alumina, Liner: HCLPE** | 0.91 | [0.81, 1.01] | 0.098 | 0.92 | [0.82, 1.03] | 0.143 | 0.94 | [0.82, 1.08] | 0.271 |
| **Head: Alumina, Liner: non-HCLPE** | 1.33 | [1.11, 1.59] | 0.003 | 1.42 | [1.21, 1.68] | <0.001 | 1.60 | [1.32, 1.93] | <0.001 |
| **Head: Delta Ceramic, Liner: Delta Ceramic** | 0.91 | [0.86, 0.97] | 0.004 | 0.93 | [0.88, 0.99] | 0.022 | 0.96 | [0.90, 1.03] | 0.197 |
| **Head: Delta Ceramic, Liner: HCLPE** | 0.78 | [0.73, 0.84] | <0.001 | 0.79 | [0.73, 0.84] | <0.001 | 0.79 | [0.73, 0.85] | <0.001 |
| **Head: Delta Ceramic, Liner: non-HCLPE** | 0.82 | [0.73, 0.92] | 0.001 | 0.84 | [0.75, 0.94] | 0.004 | 0.87 | [0.77, 0.99] | 0.038 |
| **Head: Oxidised Zirconium, Liner: HCLPE** | 0.67 | [0.58, 0.77] | <0.001 | 0.66 | [0.57, 0.77] | <0.001 | 0.65 | [0.55, 0.77] | <0.001 |
| **Head: Cobalt Chrome, Liner: non-HCLPE** | 1.30 | [1.22, 1.39] | <0.001 | 1.35 | [1.26, 1.45] | <0.001 | 1.44 | [1.33, 1.55] | <0.001 |
| **Head: Stainless Steel, Liner: HCLPE** | 0.98 | [0.88, 1.09] | 0.373 | 0.98 | [0.87, 1.10] | 0.377 | 0.98 | [0.87, 1.11] | 0.378 |
| **Head: Stainless Steel, Liner: non-HCLPE** | 1.40 | [1.26, 1.55] | <0.001 | 1.48 | [1.34, 1.64] | <0.001 | 1.62 | [1.44, 1.82] | <0.001 |
| **Head: Delta Ceramic, Pre-assembled implant** | 0.52 | [0.40, 0.68] | <0.001 | 0.53 | [0.40, 0.70] | <0.001 | 0.54 | [0.39, 0.73] | <0.001 |
| **Head: Cobalt Chrome or Stainless Steel, Pre-assembled implant** | 1.23 | [0.90, 1.67] | 0.169 | 1.20 | [0.87, 1.66] | 0.216 | 1.17 | [0.83, 1.64] | 0.265 |

Flexible parametric survival model adjusted for year of primary surgery, patient gender, age, BMI, ASA grade, implant fixation, shell composition, stem composition and head size. HCLPE, highly crosslinked polyethylene.

* Wald test.

HCLPE liner, the risks of revision for most indications were lower for implants with a delta ceramic or oxidised zirconium head with HCLPE liner.

No difference in the risk of revision could be identified between implants with oxidised zirconium head and HCLPE liner and implants with delta ceramic head and HCLPE liner (Fig R in S1 Text).

## Discussion

Compared to implants with cobalt chrome heads and HCLPE cups, the all-cause risk of revision for monobloc acetabular component primary THRs was lower for patients with a delta ceramic head and HCLPE cup implant combination. These risks were generally higher with cobalt chrome or stainless steel heads used with non-HCLPE cups. For modular acetabular components the all-cause revision risk was markedly lower when delta ceramic heads or oxidised zirconium heads were used with HCLPE. Higher risks of revision were seen with alumina heads and liners or non-HCLPE liners and for cobalt chrome and stainless steel heads with non-HCLPE liners. Similar conclusions were found when investigating indication-specific risks of revision.

The risk of revision associated with specific bearing combinations has mostly been investigated through routine reporting in registry reports without adjustment for factors that influence the risk of revision or in studies with small sample sizes that lacks the external validity of the current analysis. In a network meta-analysis of 3,177 hip replacements comparing implant survivorship by head-size, fixation and bearing, no combination had better outcomes than the reference combination of metal-on-polyethylene (not highly cross linked), small head, and cemented [25]. Due to small numbers in individual nodes the confidence intervals were very wide. As with other recent systematic reviews of RCTs considering revision outcomes with different bearing surfaces [26–30], no attempt was made to compare specific ceramic or polyethylene material combinations. From a systematic search of MEDLINE and Embase on 1 March 2023, recent registry and cohort comparative studies focusing on aspects of bearings and subsequent revision rates have not considered specific bearings surface materials including polyethylene materials or a comprehensive range of revision outcomes (Table AA in S1 Text). Consistent with our findings, in an Italian registry cohort reporting revisions due to dislocation up to 7 years after hip replacement, rates were higher with non-HCLPE liners but did not differ between different femoral head material and HCLPE combinations [31]. The low rate of revision for implant with a ceramic or oxidised zirconium head has also been reported in the Dutch Arthroplasty Register, without details by type of ceramics materials used [32].

The analysis reported here is based on an exhaustive large, nonselective data set registry, using all procedures performed by all orthopaedic units of an entire healthcare system. While several studies reported that hip implant with an alumina head had good wear properties, revision and patient reported outcomes [33–41], our results show that these implants are associated with a higher risk of revision than the most commonly used implant combinations. This had previously been reported in other studies [42,43]. The higher risk of revision for implants with a cobalt chrome head and non-HCLPE liner or cup identified in our study has also previously been reported in smaller uncontrolled studies [44–47], and in THRs using stainless steel heads [48]. Oxidised zirconium head material and HCLPE liners have also been shown to provide good results [49]. Another study could not identify any difference when compared to cobalt chrome heads [50]; our study showing good results for implant with an oxidised zirconium head provides further evidence on this bearing material combination. Two RCTs investigating oxidised zirconium heads have not reported differences in survivorship rate in comparison to cobalt chrome heads, but they were not powered to investigate revision

outcomes [50,51]. The existing observational data on THRs using delta ceramic components show positive outcomes [52–59]. Our results confirm the lower risk of all-cause and indication-specific revision for primary THR with a delta ceramic head, especially with an HCLPE liner or cup. National registries report similar lower unadjusted risk of revision for implants with ceramic head and HCLPE liner/cup compared to implants with metal head and HCLPE liner/cup [15–19]; a registry showing no evidence of difference between those 2 types of implants when the risks are adjusted for year of primary surgery, patient age, gender, and surgical year, does not report continuous longitudinal postoperative risk change as done here where we also have a larger sample size [20]. The NJR does not capture squeaking or noise as a specific indication for revision (this would be captured under other indications for revision and reflected in the all-cause revision rate if bothersome enough to lead to revision) but this is a complication that has been reported with ceramic materials in THR [60].

The mechanisms by which bearing materials might influence implant longevity are multifaceted. For the acetabular component, HCLPE has been demonstrated to have significantly reduced wear compared to non-HCLPE, reducing late failure by mitigation of particle-induced periprosthetic osteolysis which can lead to implant instability and pain [61,62]. Ceramic materials have been shown in vitro to have reduced bacterial adhesion and slower biofilm development compared to metals due to their surface properties [63]. This could influence earlier revision rates both for infection and other causes as low-grade infection can masquerade as aseptic loosening and instability [8,9]. Use of a ceramic head reduces corrosion at the trunnion (the modular interface between the head and the stem), which can cause adverse reaction to metal debris requiring revision surgery [64].

In 2022, the annual NJR reported that hip implants with a metal head, mostly in cobalt chrome, and polyethylene liner/cup had been used in 47,180 procedures and were the most used implant materials over the last 5 years [1]. Our analysis, the largest of its kind to date in a comprehensive registry providing generalisable outcome data, identifies implant materials that are associated with lower risk of revision following primary THR. For monobloc acetabular component primary THRs, implants with a delta ceramic head and HCLPE cup have the lowest risk of revision and this is sustained throughout the post-operation periods. Similarly, for modular acetabular component primary THRs, implants with a delta ceramic or oxidised zirconium heads and HCLPE liner have the lowest risk of all-cause and cause-specific revision.

These results are generalisable as they were derived from all procedures performed in England and Wales since 2003. They capture the whole diversity of clinical practices across NHS and the private sector, over an extended period. This has allowed comprehensive and nonselective comparisons between implant bearing surface materials. The modelling strategy reporting the risk of revision throughout the whole postoperative period, nearly 15 years, has allowed us to depict the time-specific risk associated with each material throughout the post-operation period by specific indication for revision. This is of particular importance given that failure rates for different indications vary over time. However, these results could be influenced by residual confounding as not all factors that could influence implant selection or the risk of revision are captured in the NJR. This is not a randomised controlled trial (RCT), and it is possible that the choice of implant materials is influenced by operating unit preference. In the United Kingdom, surgeon choice is heavily constrained at unit level due to units carrying a selected range of implants imposed by hospital board regulations, mostly for economic reasons. The indication-specific revision analyses (Figs B–Q in S1 Text) is giving further insight into the role of each bearing surface materials on the considered outcome and were aligned with the overall revision analyses (Figs 1 and 2). We cannot rule out that some of the effects identified could be partially related to selection effect. Shared operative strategies, or bearing surface selection by surgeons, within the same surgical unit, may generate some clustering at

unit level. Previous investigations on the same dataset accounting for unit level clustering added little value to the analyses and their modelling is currently challenging with large data set in the context of survival analysis [59]. Oxidised zirconium heads are only made by one manufacturer and hence are used with a small number of implant combinations which may restrict the generalisability but the group size is large and the NJR annual report shows similar results for this manufacturer's implants in comparison to others [1].

The risk of revision following primary THR is influenced by the type of material used in the bearing surface. The all-cause and indication-specific risk of revision is lower for implants with a delta ceramic head and HCLPE cups or delta ceramic heads or oxidised zirconium heads and HCLPE liners. Further work is required to determine the association of implant bearing materials with the risk of rehospitalisation, re-operation other than revision, mortality, and the cost-effectiveness of these materials.

## Supporting information

**S1 STROBE Checklist. Checklist of items that should be included in reports of cohort studies.**
(DOCX)

**S1 Text. Including: Table A.** Description of the studied sample and revision rate for aseptic loosening. **Table B.** Description of the studied sample and revision rate for peri-prosthetic fracture. **Table C.** Description of the studied sample and revision rate for implant wear. **Table D.** Description of the studied sample and revision rate for malalignment. **Table E.** Description of the studied sample and revision rate for dislocation or subluxation. **Table F.** Description of the studied sample and revision rate for pain. **Table G.** Description of the studied sample and revision rate for infection. **Table H.** Description of the studied sample and revision rate for any other reason(s). **Table I.** Monobloc acetabular component—all-cause revision hazard ratio (HR) and 95% confidence interval (CI) by time point from primary procedure-Reference: Implant with cobalt chrome head and highly crosslinked polyethylene cup. **Table J.** Monobloc acetabular component-Revision for aseptic loosening hazard ratio (HR) and 95% confidence interval (CI) by time point from primary procedure-Reference: Implant with cobalt chrome head and highly crosslinked polyethylene cup. **Table K.** Monobloc acetabular component-Revision for peri-prosthetic fracture hazard ratio (HR) and 95% confidence interval (CI) by time point from primary procedure-Reference: Implant with cobalt chrome head and highly crosslinked polyethylene cup. **Table L.** Monobloc acetabular component-Revision for implant wear hazard ratio (HR) and 95% confidence interval (CI) by time point from primary procedure-Reference: Implant with cobalt chrome head and highly crosslinked polyethylene cup. **Table M.** Monobloc acetabular component-Revision for malalignment hazard ratio (HR) and 95% confidence interval (CI) by time point from primary procedure-Reference: Implant with cobalt chrome head and highly crosslinked polyethylene cup. **Table N.** Monobloc acetabular component-Revision for dislocation or subluxation hazard ratio (HR) and 95% confidence interval (CI) by time point from primary procedure-Reference: Implant with cobalt chrome head and highly crosslinked polyethylene cup. **Table O.** Monobloc acetabular component-Revision for pain hazard ratio (HR) and 95% confidence interval (CI) by time point from primary procedure-Reference: Implant with cobalt chrome head and highly crosslinked polyethylene cup. **Table P.** Monobloc acetabular component-Revision for infection hazard ratio (HR) and 95% confidence interval (CI) by time point from primary procedure-Reference: Implant with cobalt chrome head and highly crosslinked polyethylene cup. **Table Q.** Monobloc acetabular component-Revision for any other reason(s) hazard ratio (HR) and 95% confidence interval (CI) by time point from primary procedure-Reference: Implant

with cobalt chrome head and highly crosslinked polyethylene cup. **Table R.** Modular acetabular component—all-cause revision hazard ratio (HR) and 95% confidence interval (CI) by time point from primary procedure-Reference: Implant with cobalt chrome head and highly crosslinked polyethylene liner. **Table S.** Modular acetabular component-Revision for aseptic loosening hazard ratio (HR) and 95% confidence interval (CI) by time point from primary procedure-Reference: Implant with cobalt chrome head and highly crosslinked polyethylene liner. **Table T.** Modular acetabular component-Revision for peri-prosthetic fracture hazard ratio (HR) and 95% confidence interval (CI) by time point from primary procedure-Reference: Implant with cobalt chrome head and highly crosslinked polyethylene liner. **Table U.** Modular acetabular component-Revision for implant wear hazard ratio (HR) and 95% confidence interval (CI) by time point from primary procedure-Reference: Implant with cobalt chrome head and highly crosslinked polyethylene liner. **Table V.** Modular acetabular component-Revision for malalignment hazard ratio (HR) and 95% confidence interval (CI) by time point from primary procedure-Reference: Implant with cobalt chrome head and highly crosslinked polyethylene liner. **Table W.** Modular acetabular component-Revision for dislocation or subluxation hazard ratio (HR) and 95% confidence interval (CI) by time point from primary procedure-Reference: Implant with cobalt chrome head and highly crosslinked polyethylene liner. **Table X.** Modular acetabular component-Revision for pain hazard ratio (HR) and 95% confidence interval (CI) by time point from primary procedure-Reference: Implant with cobalt chrome head and highly crosslinked polyethylene liner. **Table Y.** Modular acetabular component-Revision for infection hazard ratio (HR) and 95% confidence interval (CI) by time point from primary procedure-Reference: Implant with cobalt chrome head and highly crosslinked polyethylene liner. **Table Z.** Modular acetabular component-Revision for any other reason(s) hazard ratio (HR) and 95% confidence interval (CI) by time point from primary procedure-Reference: Implant with cobalt chrome head and highly crosslinked polyethylene liner.

**Table AA.** Registry studies identified in MEDLINE and Embase search on 1 March 2023. **Fig A.** PRISMA flow diagram. **Fig B.** Risk of revision for aseptic loosening by head and cup types (Reference: Cobalt chrome head with highly crosslinked polyethylene cup). **Fig C.** Risk of revision for aseptic loosening by head and liner types (Reference: Cobalt chrome head with highly crosslinked polyethylene liner). **Fig D.** Risk of revision for periprosthetic fracture by head and cup types (Reference: Cobalt chrome head with highly crosslinked polyethylene cup). **Fig E.** Risk of revision for periprosthetic fracture by head and liner types (Reference: Cobalt chrome head with highly crosslinked polyethylene liner). **Fig F.** Risk of revision for implant wear by head and cup types (Reference: Cobalt chrome head with highly crosslinked polyethylene cup). **Fig G.** Risk of revision for implant wear by head and liner types (Reference: Cobalt chrome head with highly crosslinked polyethylene liner). **Fig H.** Risk of revision for malignment by head and cup types (Reference: Cobalt chrome head with highly crosslinked polyethylene cup). **Fig I.** Risk of revision for malignment by head and liner types (Reference: Cobalt chrome head with highly crosslinked polyethylene liner). **Fig J.** Risk of revision for dislocation or subluxation by head and cup types (Reference: Cobalt chrome head with highly crosslinked polyethylene cup). **Fig K.** Risk of revision for dislocation or subluxation by head and liner types (Reference: Cobalt chrome head with highly crosslinked polyethylene liner). **Fig L.** Risk of revision for pain by head and cup types (Reference: Cobalt chrome head with highly crosslinked polyethylene cup). **Fig M.** Risk of revision for pain by head and liner types (Reference: Cobalt chrome head with highly crosslinked polyethylene liner). **Fig N.** Risk of revision for infection by head and cup types (Reference: Cobalt chrome head with highly crosslinked polyethylene cup). **Fig O.** Risk of revision for infection by head and liner types (Reference: Cobalt chrome head with highly crosslinked polyethylene liner). **Fig P.** Risk of revision for any other reason(s) by head and cup types (Reference: Cobalt chrome head with highly crosslinked

polyethylene cup). **Fig Q.** Risk of revision for any other reason(s) by head and liner types (Reference: Cobalt chrome head with highly crosslinked polyethylene liner). **Fig R.** Modular acetabular component-all-cause risk of revision for implants with delta ceramic head and highly crosslinked polyethylenE (HCLPE) and oxidised zirconium head and HCPLE liner.
(DOCX)

## Acknowledgments

We would like to thank Tim Wilton for his project and data oversight for this work. We thank the patients and staff of all the hospitals who have contributed data to the National Joint Registry, and the Healthcare Quality Improvement Partnership, the National Joint Registry Steering Committee, and staff at the National Joint Registry for facilitating this work.

## Author Contributions

**Conceptualization:** Michael R. Whitehouse, Elsa M. R. Marques, Ashley W. Blom, Erik Lenguerrand.

**Data curation:** Erik Lenguerrand.

**Formal analysis:** Rita Patel, Erik Lenguerrand.

**Funding acquisition:** Michael R. Whitehouse, Elsa M. R. Marques, Ashley W. Blom, Erik Lenguerrand.

**Investigation:** Michael R. Whitehouse, Rita Patel, Jonathan M. R. French, Andrew D. Beswick, Ashley W. Blom, Erik Lenguerrand.

**Methodology:** Michael R. Whitehouse, Rita Patel, Ashley W. Blom, Erik Lenguerrand.

**Project administration:** Michael R. Whitehouse, Erik Lenguerrand.

**Resources:** Erik Lenguerrand.

**Supervision:** Michael R. Whitehouse, Erik Lenguerrand.

**Validation:** Rita Patel, Jonathan M. R. French, Andrew D. Beswick, Patricia Navvuga, Elsa M. R. Marques, Ashley W. Blom, Erik Lenguerrand.

**Visualization:** Ashley W. Blom, Erik Lenguerrand.

**Writing – original draft:** Michael R. Whitehouse, Rita Patel, Jonathan M. R. French, Andrew D. Beswick, Patricia Navvuga, Elsa M. R. Marques, Ashley W. Blom, Erik Lenguerrand.

**Writing – review & editing:** Michael R. Whitehouse, Rita Patel, Jonathan M. R. French, Andrew D. Beswick, Patricia Navvuga, Elsa M. R. Marques, Ashley W. Blom, Erik Lenguerrand.

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
