## [Editor Report · Decision Letter 0]

5 Mar 2024

Dear Dr Lenguerrand, 

Thank you for submitting your manuscript entitled "The association of bearing surface materials with the risk of revision following primary total hip replacement: a cohort analysis of 1,026,481 hip replacements from the National Joint Registry" for consideration by PLOS Medicine.

Your manuscript has now been evaluated by the PLOS Medicine editorial staff as well as by an academic editor with relevant expertise and I am writing to let you know that we would like to send your submission out for external peer review.

Please re-submit your manuscript by March 7th. Please let me know if you need more time.

Kind regards,

Syba Sunny MBBS MRes FRCPath

Associate Editor

PLOS Medicine

ssunny@plos.org

---

## [Decision Letter · Decision Letter 1]

3 Jun 2024

Dear Dr. Lenguerrand,

Many thanks for submitting your manuscript "The association of bearing surface materials with the risk of revision following primary total hip replacement: a cohort analysis of 1,026,481 hip replacements from the National Joint Registry” (PMEDICINE-D-24-00694R1) to PLOS Medicine. The paper has been reviewed by three subject experts and two statisticians; their comments are included below.

As you will see, the reviewers were positive about the paper in several regards. However, concerns were raised about the way in which the data was analysed, particularly with regard to statistical rigor, and also the role of the funder, which casts some doubt over the robustness of the conclusions presented. However, we remain potentially interested in the paper if these aspects can be addressed in a substantive way. As such, after discussing the paper with the editorial team, I’m pleased to invite you to revise the paper in response to the reviewers’ comments and our editorial requests (below). We plan to send the revised paper to some of all of the original reviewers*, and of course we cannot provide any guarantees at this stage regarding publication.

When you upload your revision, please include a point-by-point response that addresses all of the reviewer and editorial points, indicating the changes made in the manuscript and either an excerpt of the revised text or the location (eg: page and line number) where each change can be found. Please submit a clean version of the paper as the main article file and a version with changes marked should as a marked-up manuscript. Please also check the guidelines for revised papers at http://journals.plos.org/plosmedicine/s/revising-your-manuscript for any that apply to your paper.

We ask that you submit your revision by 24th June. However, if this deadline is not feasible, please contact me by email, and we can discuss a suitable alternative.

Please don’t hesitate to contact me directly with any questions (ssunny@plos.org). If you reply directly to this message, please be sure to ‘Reply All’ so your message comes directly to my inbox.

Kind regards,

Syba

Syba Sunny MBBS, MRes, FRCPath

Associate Editor 

PLOS Medicine

ssunny@plos.org

*Please note: If your article is accepted, you may have the opportunity to make the peer review history publicly available. The record will include editor decision letters (with reviews) and your responses to reviewer comments. If eligible, we will contact you to opt in or out.

Editorial comments:

The editorial team all agree that this is an interesting piece of work that could be very impactful, and we are grateful that you chose to submit this to PLOS Medicine. However, we agree with the reviewers’ comments and have some concerns that will need to be addressed in full before your paper can be taken forward.

1) Thank you for stating that the study was funded by CeramTec. To ensure transparency and satisfy the team that the function of the sponsor does not preclude publication, it would be useful to add some detail on what the funding was used for, specifically, in the completion of this study.

2) Please revise your statistical analyses to address the influence of confounding variables more robustly, e.g. using rigorous sensitivity analyses. Note the comments and suggestions offered by Reviewer 3, who is a statistician.

3) Please provide a description of the formal or informal processes that might go into the decision-making process when bearing materials are chosen, with an accompanying critical discussion. This would help to address any concerns that readers might have about confounding by indication here. In your discussion, it would be useful to hear what factors specifically could have influenced decision-making – could this have been influenced by patient age or the life expectancy of the implant, for example? Our readership is global, so it would also be useful to state whether patients in the UK might have any choice over the materials used in their implants.

4) Data Availability: 

The Data Availability Statement (DAS) requires some revision – please review the first sentence. Also, please describe briefly the ethical, legal, or contractual restriction that prevents you from sharing deidentified data. 

5) Reporting guidance:

Thank you for including the STROBE checklist as Supporting Information. Please add the following statement, or similar, to the Methods: "This study is reported as per the ‘Strengthening the Reporting of Observational Studies in Epidemiology (STROBE): guidelines for reporting observational studies’ Statement” and refer to the checklist’s location.

Further guidance can be found here: https://www.equator-network.org/reporting-guidelines/

6) Statistical reporting:

Please quantify the main results with 95% CIs and p values.

When reporting p values please report as <0.001 and where higher as p=0.002, for example. When reporting 95% CIs please separate upper and lower bounds with commas instead of hyphens as the latter can be confused with reporting of negative values.

Please include the actual amounts and/or absolute risk(s) of relevant outcomes (including NNT or NNH where appropriate), not just relative risks or correlation coefficients. (example for absolute risks: PMID: 28399126).

Please include any important dependent variables that are adjusted for in the analyses.

7) Prespecified analysis plan/study protocol:

Did your study have a prospective protocol or analysis plan? Please state this (either way) early in the Methods section.

For all observational studies, in the manuscript text, please indicate: (i) the specific hypotheses you intended to test, (ii) the analytical methods by which you planned to test them, (iii) the analyses you actually performed, and (iv) when reported analyses differ from those that were planned, transparent explanations for differences that affect the reliability of the study's results. If a reported analysis was performed based on an interesting but unanticipated pattern in the data, please be clear that the analysis was data-driven.

8) Author summary:

At this stage, we ask that you include a short, non-technical Author Summary of your research to make findings accessible to a wide audience that includes both scientists and non-scientists. The authors summary should consist of 2-3 succinct bullet points under each of the following headings:

• Why Was This Study Done? Authors should reflect on what was known about the topic before the research was published and why the research was needed.

• What Did the Researchers Do and Find? Authors should briefly describe the study design that was used and the study’s major findings. Do include the headline numbers from the study, such as the sample size and key findings. 

• What Do These Findings Mean? Authors should reflect on the new knowledge generated by the research and the implications for practice, research, policy, or public health. Authors should also consider how the interpretation of the study’s findings may be affected by the study limitations. In the final bullet point of ‘What Do These Findings Mean?’, please describe the main limitations of the study in non-technical language.

The Author Summary should immediately follow the Abstract in your revised manuscript. This text is subject to editorial change and should be distinct from the scientific abstract. Please see our author guidelines for more information: https://journals.plos.org/plosmedicine/s/revising-your-manuscript#loc-author-summary

Comments from the reviewers:

REVIEWER #1: 

This study investigated the revision rate of primary THR reported in the National Joint Registry by specific types of bearing surfaces used.

Comments:

"We assessed data from the National Joint Registry (NJR)—established in 2003. It currently records all primary and revision hip replacements done in hospitals in England, Wales, Northern Ireland, the Isle of Man and the States of Guernsey. A total of 1,204,423 primary THR procedures had been recorded in England and Wales until 31st December 2019. Our analysis is based on 1,026,481 (85.2%) primary procedures, recorded in the NJR, which include patient consent and identifiers that allow revisions to be linked to primary operations with an identifiable head-cup or head-liner combination. "

A wealth of valuable and representative data has been used.

"unit of analysis for the consideration of revision outcomes is the implant (rather than patient) so we included 260,830 primary procedures performed on contralateral sides of the same patient but on different dates. "

Clustering by patient considered?

"used flexible parametric survival models that estimate hazard ratios (HR) by bearing materials assuming that they were likely to vary over time.(20) Restricted cubic spline function used to model the baseline function and the time-dependent effects associated with bearing materials combination only were determined using the smallest Akaike Information Criterion (AIC) and Bayesian Information Criterion (BIC).(21) The analyses were adjusted for patient age, gender, BMI, ASA grade, implant fixation, head component size, and stem materials. We also assessed HRs for each specific reason for revision described above."

Appropriate statistical models, with insightful additional analyses, have been conducted.

The Results are communicated accurately and clearly, and the figures are understandable and informative.

"However, these results could be influenced by residual confounding as not all factors that could influence implant selection or the risk of revision are captured in the NJR. This is not a randomised controlled trial (RCT), and it is possible that the choice of implant materials is influenced by surgeon preference. This choice may be constrained at the unit level due to units carrying a selected range of implants, but we cannot rule out that some of the effects identified could be partially related to selection effect. Shared operative strategies, or bearing surface selection by surgeons, within the same surgical unit, may generate some clustering at unit level. Previous investigations on the same dataset accounting for unit level clustering added little value to the analyses and their modelling is currently challenging with large dataset in the context of survival analysis.(22) Oxidised zirconium heads are only made by one manufacturer and hence are used with a small number of implant combinations which may restrict the generalisability but the group size is large and the NJR annual report shows similar results for this manufacturer's implants in comparison to others."

The main study limitations have been acknowledged in the discussion.

REVIEWER #2: 

"Restricted cubic spline function used to model the baseline function" - suggest changing "baseline function" to "baseline hazard" for clarity

"the time-dependent effects associated with bearing materials combination only were determined using the smallest Akaike Information Criterion (AIC) and Bayesian Information Criterion (BIC)." - I do not completely understand what you mean here - did you investigate different forms of the restricted cubic splines (e.g., number/placement of knots) and then use AIC/BIC to determine best fit? 

Did you consider adjusting for year of primary THA? It would seem that trends in implant type would have changed from 2003 - 2019, with some implant types more likely to be used in earlier years and other in later years. 

How did the models account for the clustered nature of the data (potential for multiple implants per patient)? 

Did the analyses for specific revision reasons include the competing risk of other revision reasons? 

REVIEWER#3: 

This is an interesting study on the association of bearing surface materials with the risk of revision following primary total hip replacement using data from a National Joint Registry. However, there are a few major issues needing attention.

1. The paper is difficult to follow especially for non-expert in the hip replacement field. With so many sub-categories and combinations between them (table 1), it is easy to get lost as it's not clear why the reference was chosen and why those comparisons were chosen (table 2).

2. Study design. Ideally RCTs are perferred to cohort studies for this type of study as there are many confounding factors assiciated the risk of revision following THR. Unless adjusted comprehensively for all the potential confounders, the results are subject to scrutiny. For example, year of hip replacement, ethnicity, socioeconomic status, drinking, smoking, multimorbidity, hip replacement procedures, medication and treatment in the follow-ups, all these factors could contribute to the revision but unfortunately not included in the analyses.

3. Statistical analyses. Flexible parametric survival models were used to reflect the time-dependant nature of HRs, which is mostly adequate. However, as the outcome is revision rather than all-cause mortality, there is a competing risk issue (eg. from death or not qualified for a revision due to deteriorating health) which was not addressed in the paper.

4. The abstract was poorly written. There is no HRs or P-values in the findings when higher or lower was claimed. As there were so many comparisons, it became difficult to follow what the findings are, what the conclusions are and what the interpretations are. Basically, it is difficult to find and understand key messages from the paper.

REVIEWER #4: 

Thank you for letting me review this interesting paper. The authors have compared the risk of revision between different articulations in a national registry setting. Extensive statistical analyses have been carried out. The paper is well written, the aim is well described, the subject is timely and the material is substantial. Main findings were that cross-linked PE was better than non-crosslinked PE, delta ceramic and Oxinium was superior to CoCr, and so was Alumina and stainless steel. Differences were found for 'all revisions' and for 'indication-specific revisions' and were mostly observed in the first years post implantation.

The manuscript leaves me with some questions: Is a difference in dislocation, loosening, fracture or infection rate the first few years after THR caused by the articulating materials? From a clinical point of view, I would be surprised to find that the materials were responsible. In this paper no possible explanations for the findings are discussed. The reader is left with the conclusion that ceramics lower the risk of early dislocation, infection and so on. This could of course be the case, but other factors, such as time dependent factors that may or may not be adjusted for, could also affect the findings. Could poor stem or cup makes be distributed disproportionally between the groups? Differences in median follow-up could affect the risk of revision for infection, and maybe other causes also. I believe this should be explored. A critical discussion of findings that are not easily explainable from a clinical point of view would be appreciated. This is maybe particularly important because the study is sponsored by the biggest player in the ceramic industry. I suggest doing some kind of sensitivity analysis, for instance the authors may try to study a subset of patients operated in the same time period, with the same implants, in similar patients (i.e. OA, ASA 1-2, same age distribution) but with different articulating surfaces. If the findings are replicated in such sub-analyses this would certainly strengthen the results.

Page 4, line 11. Needs a reference

Page 4, line 15. Please see NARA study. Mikkelsen RT et al, Acta Orthop 2023

Page 5, line 3-7. See Mikkelsen study

Page 7, line. I am a bit confused about the pre-assembled cups; where they grouped with the non-HCLPE liners as stated in line 3, 4?

Page 7, lines 15-17. Some factors tend to vary with time over two decades, for instance the risk of revision for infection. Did you consider adjusting for year of operation or stratified time periods? What about make of cup and stem? Poorly performing implants could skew the results, and such implants were maybe more frequent in the beginning of the study period?

Table 1. DM cups were excluded. Still some 7-800 modular cups were cemented. What were these?

Page 18, Line 23-25. Reference 31. The authors studied fractures, they did not report on dislocations.

Also I miss information on median follow-up for the study groups.

I am not familiar with all the tests that were used. It seems that the risks varied with time, and I do not know if the chosen tests handle non-proportionality properly. A statistical review will probably be called for.

REVIEWER#5: 

This research study reports a cohort analysis of hip replacements reported in the UK NJR about the association of bearing surfaces materials with the risk of revision following primary total hip replacement. It is a very interesting topic of research since it contributes significantly to understanding the best option for the tribology of primary hip replacements. Moreover, this paper takes into consideration a remarkable amount of data that can be easily generalized. However, some minor changes are needed to be suitable for pubblication.

Introduction. This section can be shortened and focused more on the topic of research, that is the relationship between the tribology of the prosthesis and the risk of revision. Some lines do not give added value to the topic (ex: 3-4, 9-11, 21-23).

M&M:

- Data source, line 6: numbers given are in conflict with what is described in tables. The total amount of replacements analysed according to tables is 1.027.098, monoblocs are 378.897 and modulars are 648.201. A comprehensive check of the numbers reported should be done throughout the manuscript.

- I was not able to find eFigure 1

- How was the reference group (cobalt chrome head + HCLPE cup) determined? There is no mention of the reference group in this section

Results:

- Tables are difficult to follow. Highlighting significant positive results (lower risk of revision for "experimental groups") could be of help

- "This lower revision rate was limited to the first year when a delta ceramic head 7 was paired with a delta ceramic or non-HCLPE liner" (page 14, lines 6-7), according to sTable18, lower revision rate is seen at least up to 2 years from surgery

- Page 14, lines 16-18. Heads and cups combinations other than the reference group have been compared. However, there is no mention in the text to the reason of this choice. 

Discussion. This section is difficult to follow: a revision of the structure should be done. It could be better to summarize the results at first, then compare them to previous work already available in the Literature and at the end higlight limitations and strenght of the study. Moreover, some lines (17-19) should be included in M&M section rather than in the discussion.

[LINK]

1. Please upload any figures associated with your paper as individual TIF or EPS files with 300dpi resolution at resubmission; please read our figure guidelines for more information on our requirements: http://journals.plos.org/plosmedicine/s/figures. While revising your submission, please upload your figure files to the PACE digital diagnostic tool, https://pacev2.apexcovantage.com/. PACE helps ensure that figures meet PLOS requirements. To use PACE, you must first register as a user. Then, login and navigate to the UPLOAD tab, where you will find detailed instructions on how to use the tool. If you encounter any issues or have any questions when using PACE, please email us at PLOSMedicine@plos.org.

To submit your revised manuscript please use the following link:

---

## [Decision Letter · Decision Letter 2]

16 Aug 2024

Dear Dr. Lenguerrand,

Thank you very much for re-submitting your manuscript "The association of bearing surface materials with the risk of revision following primary total hip replacement: a cohort analysis of 1,026,481 hip replacements from the National Joint Registry" (PMEDICINE-D-24-00694R2) for review by PLOS Medicine.

I have discussed the paper with my colleagues, the reviewers and a guest academic editor. I am pleased to say that, provided the remaining editorial and production issues are dealt with, we are planning to accept the paper for publication in the journal. The remaining issues that need to be addressed are listed at the end of this email. 

We expect to receive your revised manuscript within 1 week. Of course, please do let me know directly if you need more time (ssunny@plos.org). Please email us (plosmedicine@plos.org) if you have any questions or concerns about your submission.

We look forward to receiving the revised manuscript by Aug 23 2024 11:59PM. 

Sincerely,

Syba

Syba Sunny, MBBS, MRes, FRCPath

Associate Editor 

PLOS Medicine

ssunny@plos.org

COMMENTS FROM THE EDITORS

Firstly, thank you for your detailed and thorough responses to previous requests and comments. 

We have another reviewer request that will need to be addressed (see Reviewer 5’s comments and those from the guest academic editor below). We also have a number of editorial requests, the majority of which simply pertain to journal-specific style and formatting requirements; I have put these at the very bottom of this email.

COMMENTS FROM REVIEWERS

Reviewer #2: Thank you for the comprehensive response. I have no further questions. 

Reviewer #5: Authors have answered point-to-point to the comments and now the paper is more impactful.

I have a minor revision to suggest. It could be better to include in the manuscript the reasons of the choice of the reference group. In the answers given it was clearly stated, but there is no reference in the main text. It could be helpful to better understand the comparisons even for non-subspecialist readers.

Reviewer #6: For context, this review was taken on upon request by the Associate Editor, as a stand-in for the original stats reviewer (Reviewer #3). The major relevant points raised were on study design (RCTs vs. cohort study), and on the absence of competing risks models.

On the use of a cohort study (instead of RCT), my opinion is that cohort studies are valid, as explained by the authors (and also acknowledged by the original Reviewer #3, who describes RCTs as "preferred"). Given the difficulty of organizing RCTs on the subject, a (large) cohort study would provide a useful datapoint which would be of use in future meta-analyses (with risks and limitations properly acknowledged).

On the absence of addressing competing risks, the authors' rebuttal appears reasonable. It might be emphasized within the manuscript itself.

[LINK]

COMMENTS FROM THE GUEST ACADEMIC EDITOR

Reviewer 5 was invited to act as the academic editor for your revised manuscript. She looked through all the reviewer comments as well as your revised paper. She wrote that she believed that your paper was well-written and tackled an interesting topic for the orthopaedic community. She was keen for your paper to be taken forward to publication but asked if the authors could address the point she made as Reviewer 5, namely that you include a more include a more precise explanation regarding the reference group of comparisons in the main text, as she believes this would be beneficial for non-subspecialist readers.

EDITORIAL REQUESTS

Data Availability Statement (DAS):

Thank you for your detailed DAS. I have only 1 minor amendment to suggest; the first sentence contains the phrase ‘…the data were analysed in a data safe heaven…’ – I think this is meant to read as ‘data safe haven’. Could you revise this if appropriate please?

Reporting guidance:

Thank you for amending your STROBE checklist with reference to the sections rather than page numbers.

Please include a statement in your Methods section of your main text stating ‘This study is reported as per the ‘Strengthening the Reporting of Observational Studies in Epidemiology (STROBE): guidelines for reporting observational studies’ Statement’ and also refer to the checklist’s location.

Statistical reporting:

Thank you for revising your tables such that 95% Cis are separated by commas instead of hyphens. I note that, in some places, CIs are still reported with a hyphen and not with a comma, e.g. in the Results section of the main text. We would appreciate it if you could review your entire document to check and revise this accordingly. 

Author summary:

Thank you for providing an author summary. This will need to be revised to match the journal style, as below:

The Author Summary is a short, non-technical summary of your research to make findings accessible to a wide audience that includes both scientists and non-scientists. The authors summary should consist of 2-3 succinct bullet points under each of the following headings:

• Why Was This Study Done? Authors should reflect on what was known about the topic before the research was published and why the research was needed.

• What Did the Researchers Do and Find? Authors should briefly describe the study design that was used and the study’s major findings. Do include the headline numbers from the study, such as the sample size and key findings. 

• What Do These Findings Mean? Authors should reflect on the new knowledge generated by the research and the implications for practice, research, policy, or public health. Authors should also consider how the interpretation of the study’s findings may be affected by the study limitations. In the final bullet point of ‘What Do These Findings Mean?’, please describe the main limitations of the study in non-technical language.

Author Summary should immediately follow the Abstract in your revised manuscript. This text is subject to editorial change and should be distinct from the scientific abstract. Please see our author guidelines for more information: https://journals.plos.org/plosmedicine/s/revising-your-manuscript#loc-author-summary

---

## [Decision Letter · Decision Letter 3]

23 Sep 2024

Dear Dr Lenguerrand, 

On behalf of my colleagues and the Guest Academic Editor, I am pleased to inform you that we have agreed to publish your manuscript "The association of bearing surface materials with the risk of revision following primary total hip replacement: a cohort analysis of 1,026,481 hip replacements from the National Joint Registry" (PMEDICINE-D-24-00694R3) in PLOS Medicine.

Before your manuscript can be formally accepted you will need to address some minor editorial requests (see below) and complete some formatting changes, which you will receive in a follow up email. Please be aware that it may take several days for you to receive this email; during this time no action is required by you. Once you have received these formatting requests, please note that your manuscript will not be scheduled for publication until you have made the required changes.

PRESS

Sincerely, 

Syba

Syba Sunny, MBBS, MRes, FRCPath 

Associate Editor 

PLOS Medicine

Editorial Requests:

- In the Abstract, first sentence, could you include:

(1) ‘otherwise known as’ before the word ‘revision’

(2) ‘in part’ after the word ‘depends’

So that it reads ‘The risk of re-operation, otherwise known as revision, following primary hip replacement depends, in part, on the prosthesis implant materials used.’

- In the Abstract, third sentence, please expand THR before introducing this abbreviation, so that it reads: ‘We investigated the revision rate of primary total hip replacement (THR) reported in the National Joint Registry by specific types of bearing surfaces used.’

- Author Summary: The first sentence is a little confusing. Suggest changing it to perhaps ‘The classifications used to categorise hip implants in national registries are typically broad and may not allow interested parties to fully understand the risk of postoperative revision (i.e. need for further surgery) associated with different types of implant materials.’

- Methods section, under ‘Statistical analysis’ – thank you for including the statement ‘This study is reported as per the ‘Strengthening the Reporting of Observational Studies in Epidemiology (STROBE): guidelines for reporting observational studies’ Statement’ (See statement).’ Please replace the ‘(See statement)’ at the end of that sentence with the location of the file, i.e. where in the supporting information it can be found, as appropriate.

- Results section, under ‘Modular acetabular implants (n=647,502)’, the following sentence needs revision (possibly missing a word after the second ‘alumina’, and also has 2 full stops at the end of the sentence): ‘From 6 months post-operation onwards, the risk of revision was higher for implants with an alumina head and alumina, or non-HCLPE liner..’

- In the Discussion section, there is this sentence: ‘The NJR does not capture squeaking or noise as a specific indication for revision “(this would be captured under other indications for revision and reflected in the all-cause revision rate if bothersome enough to lead to revision)” but this is a complication that has been reported with ceramic materials in THR.(60)’ Is the bit in double quotation marks (“) supposed to be quoting correspondence? Can you revise accordingly, please?